# Combined effect of probiotics and omega-3 fatty acids on blood and liver function biomarkers and gut microbiota diversity in male mice with high fat diet-induced obesity

Kwang Hee Son[1,2☉], Jeongwook Lee[3☉], Hee-Young Yang[2], Yerim Heo[3], Jong Seo Lee[3], Dong-Seok Lee[1,4], Kum-Joo Shin[3*], Tae-Jun Kwon[2*]

**1** School of Life Sciences & Biotechnology, College of Natural Sciences, Kyungpook National University, Daegu, Republic of Korea, **2** Preclinical Research Center, Daegu-Gyeongbuk Medical Innovation Foundation (K-MEDI hub), Daegu, Republic of Korea, **3** R&D Center, Hecto Healthcare Co., Ltd., Seoul, Republic of Korea, **4** School of Life Sciences, BK21 FOUR KNU Creative BioResearch Group, Kyungpook National University, Daegu, Republic of Korea

☉ These authors contributed equally to this work.
* kjshin@hecto.co.kr (K-JS); tjkwon@kmedihub.re.kr (T-JK)

## Abstract

This study investigates the combined effects of probiotics and omega-3 fatty acids on blood and liver function biomarkers, as well as gut microbiota diversity, in a high-fat diet (HFD)-induced obesity model in male mice. Cardiovascular diseases (CVD), often linked to obesity, are major global health issues driven by abnormal cholesterol and triglyceride levels in the blood. Omega-3 fatty acids, known to lower triglycerides and prevent CVD, and probiotics, which enhance gut microbiota balance and nutrient absorption, were evaluated for their combined effects. The experiment involved five groups of male C57BL/6J mice: a normal diet group, an HFD group, an HFD + probiotics group, an HFD + omega-3 group, and an HFD + combined probiotics and omega-3 group. Over six weeks, the combined treatment group showed significant reductions in body weight gain, a significant improvement in ALT levels (a key liver function biomarker), and enhanced anticoagulation markers, such as prothrombin time and activated partial thromboplastin time, compared to the HFD group. Gut microbiota analysis via 16S rRNA sequencing also revealed significant increases in microbial diversity in the combined treatment group. These findings suggest that co-administration of probiotics and omega-3 offers potential therapeutic benefits in reducing obesity-related metabolic dysfunctions by improving lipid metabolism, liver health, and blood circulation.

**Data availability statement:** All relevant data are within the manuscript and its Supporting Information files.

**Funding:** This work was supported by the National Research Foundation of Korea(NRF) grant funded by the Korea government(MSIT) (Grant number: RS-2022-NR073650 to T.J.K). The funders had no role in study design, data collection and analysis, decision to publish, or preparation of the manuscript.

**Competing interests:** The authors have declared that no competing interests exist.

## Introduction

Cardiovascular diseases (CVD) are major causes of death worldwide, characterized by an abnormal concentration of factors such as total cholesterol and triglycerides in the blood due to excessive nutrient intake and lack of exercise, and can lead to disorders of blood circulation, such as hyperlipidemia and hypertension [1–3]. Therefore, various new drugs and dietary supplements that can improve blood circulation are being developed worldwide to prevent CVD.

Omega-3 fatty acids, such as eicosapentaenoic acid (EPA) and docosahexaenoic acid (DHA), have been shown to lower the concentration of triglycerides, a factor that can induce CVD, and exhibit efficacy in anti-coagulation and anti-venous thrombosis, contributing to the prevention of CVD [4]. Since omega-3 fatty acids can only be partly synthesized in the body, their availability largely depends on dietary intake; therefore, the absorption rate is a critical factor in determining their physiological effectiveness.

As with the study that showed that the absorption rate of calcium increases when taken with polygammaglutamic acid compared to when taken with calcium alone, it is important to discover dietary supplements that can be taken together to increase the absorption rate of omega-3 and see synergistic effects [5,6].

Probiotics are live microorganisms that have beneficial effects on the human body when consumed, improving the balance of gut microbiota, improving obesity, lowering cholesterol levels, and reducing inflammation [7–10]. Probiotics also enhance the absorption of nutrients such as vitamins, minerals, and fatty acids, showing synergistic efficacy when taken together [11–13]. Moreover, research findings have reported synergistic effects in combination probiotic strains compared to single strains in models of conditions such as colitis and anti-obesity [14,15]. Therefore, there is a need for combination studies of probiotics tailored to various diseases, as these studies have shown that a combination of multiple strains exhibits synergistic effects compared to a single strain.

In this study, we selected the De Simone formulation (DSF, composed of *bifidobacteria*, *lactobacilli*, and *Streptococcus thermophilus* strains), which is known to improve the absorption rate of various nutrients such as vitamins, fatty acids, as well as the efficacy of intestinal inflammation and immune diseases, to confirm the synergistic effect of increasing the absorption rate of omega-3 [16,17]. This study aims to evaluate the combined effects of probiotics and omega-3 co-administration in a high-fat diet-induced obesity model by analyzing blood and liver function biomarkers, as well as to assess changes in gut microbiota diversity through microbiome analysis.

## Materials and methods

### Animal and ethics statement

Five weeks old specific pathogen-free male C57BL/6J mice were purchased from The Jackson Laboratory Japan (Yokohama, Kanagawa, Japan). All animals were housed in ventilated IVC cage at temperature of $22 \pm 1°C$, humidity of $50 \pm 10\%$, and a 12-hour light/dark cycle with free access to food and water. The animal experiments

were approved by the Institutional Animal Care and Use Committee (IACUC) of Daegu-Gyeongbuk Medical Innovation Foundation (K-MEDI hub) (approval No. KMEDI-23011704–00) and were in accordance with IACUC guidelines.

## Experimental design

The experimental groups consisted of five groups (n = 10 per group): 1) normal diet (ND), 2) high fat diet (HFD), 3) HFD + DSF, 4) HFD + omega-3, and 5) HFD + DSF and omega-3 combination. The ND and HFD groups were orally administered 0.1 mL of PBS. The probiotics was administered 0.1 mL (2.5x10$^9$ CFU/mouse) of probiotics that suspended in PBS and then the omega-3 was treated 300 mg/kg based on body weight, daily for 6 weeks. The mice in the ND group were fed a normal diet (SAFE A40, SAFE Inc., Augy, France) containing 3.2% fat, whereas HFD fed mice were provided high fat diet (D12492, Research Diets Inc., New Brunswick, NJ, USA) containing 60% fat, 20% proteins, and 20% carbohydrates. During the experimental period, the mice body weight and food intake were measured twice weekly. The food efficiency ratio (FER) was calculated by dividing the weekly body weight gain by the food intake.

## Biochemical analysis

After 6 weeks of treatment, the mice were anesthetized with isoflurane (Hana Pharm, Co., Ltd., Seoul, Korea), and blood was collected. Thereafter, the mice were euthanized by cervical dislocation, and the liver was collected. The plasma was separated by centrifugation at 12,000 rpm and 4°C for 5 minutes. The plasma was stored at −70°C until analysis, and levels of aspartate aminotransferase (AST), alanine aminotransferase (ALT), total cholesterol, high-density lipoprotein cholesterol (HDL), low-density lipoprotein cholesterol (LDL), and triglyceride were measured using a clinical chemistry analyzer (TBA-120FR, Toshiba, Tokyo, Japan). Total cholesterol and triglyceride levels in the liver were measured using an ELISA kit (BIOMAX, Guri, Korea), following the manufacturer's instructions. Liver tissues were first homogenized in an appropriate buffer to extract lipids. The homogenized samples were then subjected to the ELISA procedure, and the resulting reactions were quantified by measuring the optical density (OD) at 570 nm using a microplate reader (Tecan Spark ®, Tecan Trading AG, Zurich, Switzerland).

## Assessment of prothrombin time (PT) and activated partial thromboplastin time (aPTT)

The anticoagulant activity of the intrinsic blood coagulation pathway was measured by prothrombin time (PT) and activated partial thromboplastin time (aPTT) in the plasma using CA-660 (SYSMEX, Milton Keynes, UK). Plasma was obtained by centrifugation after mixing with 3.2% sodium citrate for PT and aPTT measurement.

## Genomic DNA extraction of faecal samples

Fecal samples collected at the end of the six-week treatment period were subjected to 16S rRNA gene sequencing to profile the gut microbial communities. The samples were then stored at −80°C until DNA extraction. The total bacterial genomic DNA extraction from fecal samples were carried out using a Maxwell® RSC PureFood GMO and Authentication Kit (Promega, Madison, WI, USA), according to the manufacturer's instructions. The concentration of the DNA was determined by means of a UV-vis spectrophotometer NanoDrop 2000c (Thermo Scientific™, Waltham, USA), while the quantification of DNA was performed using a QuantiFluor® ONE dsDNA System (Promega, Madison, WI, USA). All extracted DNA samples were stored at –20°C until their use in further experiments.

## Gut microbiota analysis based on faecal samples using NGS

The V3-V4 variable region of the 16S rRNA gene was amplified from DNA extracts using the 16S metagenomic sequencing library protocol (Illumina, San Diego, USA). Two PCR reactions were completed on the template DNA. Initially the DNA was amplified with primers specific to the V3-V4 region of the 16S rRNA gene which also incorporates the Illumina

overhang adaptor (Forward primer 5' TCGTCGGCAGCGTAGGATGTGTATAAGAGACAGCCTACGGGNGCWGCAG; Reverse primer 5' GTCTCGTGGGCTCGGAGATGTGTATAAGAGACAGGACTACHVGGGTA-TCTAATCC). Each PCR reaction contained DNA template (~10–12ng), 5 µl forward primer (1 µM), 5 µl reverse primer (1 µM), 12.5 µl 2X Kapa HiFi Hotstart ready mix (Kapa Biosystems, Wilmington, USA), PCR grade water to a final volume of 25µl. PCR amplification was carried out as follows: heated lid 110°C, 95°C x 3 mins, 25 cycles of 95°C x 30s, 55°C x 30s, 72°C x 30s, and then 72°C x 5mins and held at 4°C. PCR products were visualized using gel electrophoresis (1X TAE buffer, 2% agarose, 100V). Successful PCR products were cleaned using AMPure XP magnetic bead based purification (Beckman Coulter, Brea, UK) and run on the Agilent Bioanalyzer for quality analysis. A second PCR reaction was completed on the purified DNA (5µl) to index each of the samples. Two indexing primers (Illumina Nextera XT indexing primers, Illumina, Sweden) were used per sample. Each PCR reaction contained 5µl index 1 primer (N7xx), 5µl index 2 primer (S5xx), 25µl 2x Kapa HiFi Hot Start Ready mix, 10µl PCR grade water. PCRs were completed as described above, but only 8 amplification cycles were completed instead of 25. Successful PCR products were cleaned using AMPure XP magnetic bead based purification (Beckman Coulter, Brea, UK). The purified products were quantified using a QuantiFluor® ONE dsDNA System (Promega, Madison, WI, USA) and run on the Agilent Bioanalyzer. The sample pool (4nM) was denatured with 0.2N NaOH, then diluted to 20pM and combined with 10% (v/v) denatured 8pM PhiX, prepared following Illumina guidelines. Samples were sequenced on the MiSeq sequencing platform, using a 2 x 300 cycle V3 kit, following standard Illumina sequencing protocols.

## 16S rRNA sequencing data analysis

The gut microbiota analysis was performed with QIIME 2 2023.05 pipeline [18]. Paired end sequence data were demultiplexed using MiSeq Reporter and joined using the q2-vsearch plugin. Merged sequences were quality filtered using the q2-quality-filter plugin followed by denoising with Deblur [19] (via q2-deblur). All amplicon sequence variants (ASVs) were aligned with mafft [20] (via q2-alignment). Taxonomy was assigned to ASVs using the q2-feature-classifier [21] classify-sklearn naïve Bayes taxonomy classifier against the SILVA DB v138.1 [22]. ASVs were identified from phylum to genus level. Non-bacterial sequences (including archaea, mitochondria, chloroplasts, and eukaryotic contaminants) were removed. To address variability in sequencing depth, rarefaction was performed using the sample with the lowest read count. The relative abundance of each taxon was calculated as the proportion of total reads within each sample.

Diversity metrics were calculated to assess both within-sample (alpha diversity) and between-sample (beta diversity) variation. Alpha diversity metrics (Shannon) and beta diversity metrix (Bray-Curtis dissimilarity), and Principle Coordinate Analysis (PCoA) were estimated using q2-diversity after samples were rarefied (subsampled without replacement). Permutational multivariate analysis of variance (PERMANOVA) with 10,000 repeated measures was performed to assess statistical differences in microbial community composition between groups [23]. We carried out linear discriminant analysis effect size (LEfSe) analysis [24] to detect significant differences in bacterial taxonomies (LDA score > 3.0).

## Statistical analysis

Statistical analysis of animal study was performed using Prism 10 software (GraphPad Software Inc., San Diego, CA, USA). For comparisons of more than two groups, a one-way or two-way ANOVA was performed, followed by a post-hoc test with Dunnett's correction for pairwise group differences. All data are expressed as the mean±SD (standard deviation) and $p < 0.05$ was considered statistically significant. All data visualization and statistical analyses of microbiome were performed using R (version 4.3.3), with visualization primarily using ggplot2 (3.5.1), and statistical tests using the package rstatix (0.7.2). After Kruskal Wallis test for multiple comparisons among groups, by index or taxon, if there was a significant index or taxon, Dunn's test was performed as a post hoc test.

## Results

### Probiotics and omega-3 combined treatment inhibited body weight gain

To examine the effects of probiotics and omega-3, we measured body weight, food intake, and food efficiency twice a week for a total of 6 weeks. The body weight gain of the mice in the HFD group was significantly higher compared to the mice in the ND group at 6 weeks after treatment (p < 0.001). At 21 days after treatment, the mice in the HFD + omega3 and HFD+DSF + Omega3 groups showed a significant reduction in body weight gain compared to the HFD group (Fig 1A). After 6 weeks of treatment, the HFD + DSF + Omega3 group showed a decrease in body weight gain of about 8% compared to the HFD group (Fig 1B-C).

### Probiotics and omega-3 combined treatment improved liver function

In this study, we aimed to investigate the effects of probiotics and omega-3 supplementation on liver function in a HFD mouse model. We analyzed total cholesterol, triglycerides levels and AST, ALT levels in the liver and plasma, respectively. Compared to the HFD group, which measured 0.25 ± 0.05 µg in total cholesterol, the HFD + DSF and HFD + omega3 groups showed no significant difference, and then the HFD + DSF + Omega3 group showed a value approximately 15% lower (0.21 ± 0.07 µg). In addition, when compared to the ND group, the HFD, HFD + DSF, and HFD + omega3 groups showed a significant difference, while the HFD + DSF + Omega3 group did not show a significant difference (Fig 2A). Regarding triglycerides levels, the HFD, HFD + DSF, HFD + omega3, and HFD + DSF + Omega3 groups showed 0.77 ± 0.17 nmol, 0.46 ± 0.08 nmol, 0.49 ± 0.17 nmol, and 0.56 ± 0.16 nmol, respectively, and statistical significance was observed in all groups compared to HFD group, but no additional effect of combined probiotics and omega3 treatment was observed (Fig 2B). Regarding AST and ALT levels, the HFD + Co-treat group showed a significant decrease in ALT levels compared to the HFD group, while AST levels showed a non-significant decreasing trend (Fig 2C-D).

### Probiotics and omega-3 combined treatment reduced lipid dyslipidemia factors

We measured the plasma levels of total cholesterol, HDL, LDL and triglyceride. At the 6 weeks, the total cholesterol in the plasma of the HFD group was 157.00 ± 7.32 mg/dL, while the HFD+omaga3 and the HFD + DSF + Omega3 groups showed 136.50 ± 12.92 mg/dL and 141.10 ± 10.86 mg/dL, respectively, and then were statistically significantly lower than the HFD group (Fig 3A). In addition, the LDL in the plasma of the HFD group was 12.08 ± 0.95 mg/dL, while the HFD+omaga3 and the HFD + DSF + Omega3 groups showed 10.85 ± 1.10 mg/dL and 10.79 ± 0.97 mg/dL, respectively, and

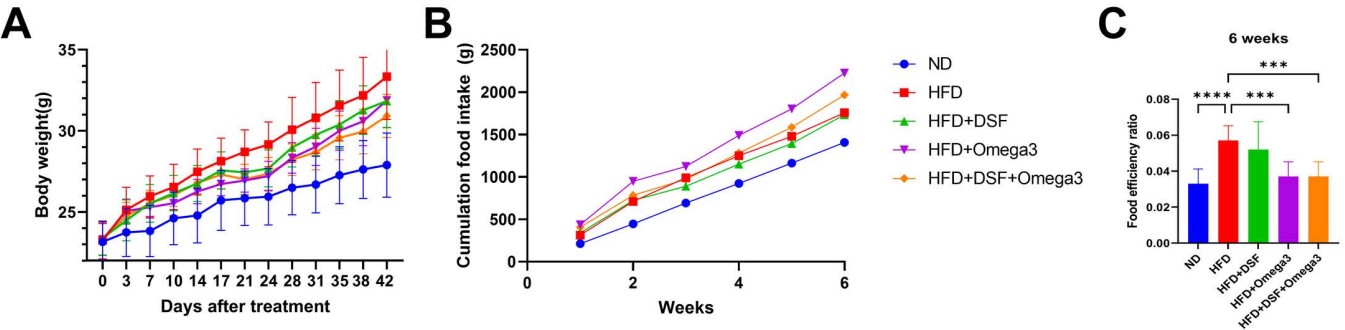

**Fig 1. Efficacy of combined administration of probiotics and omega-3 in high fat diet fed obese mice.** (A) Body weight gain in mice, (B) food intake gain, and (C) food efficiency ratio at 6 weeks. Values represent the mean ± SD (n = 10 per group). Significance is indicated by *p < 0.05, **p < 0.01, ***p < 0.001, and ****p < 0.0001 as compared with the HFD group.

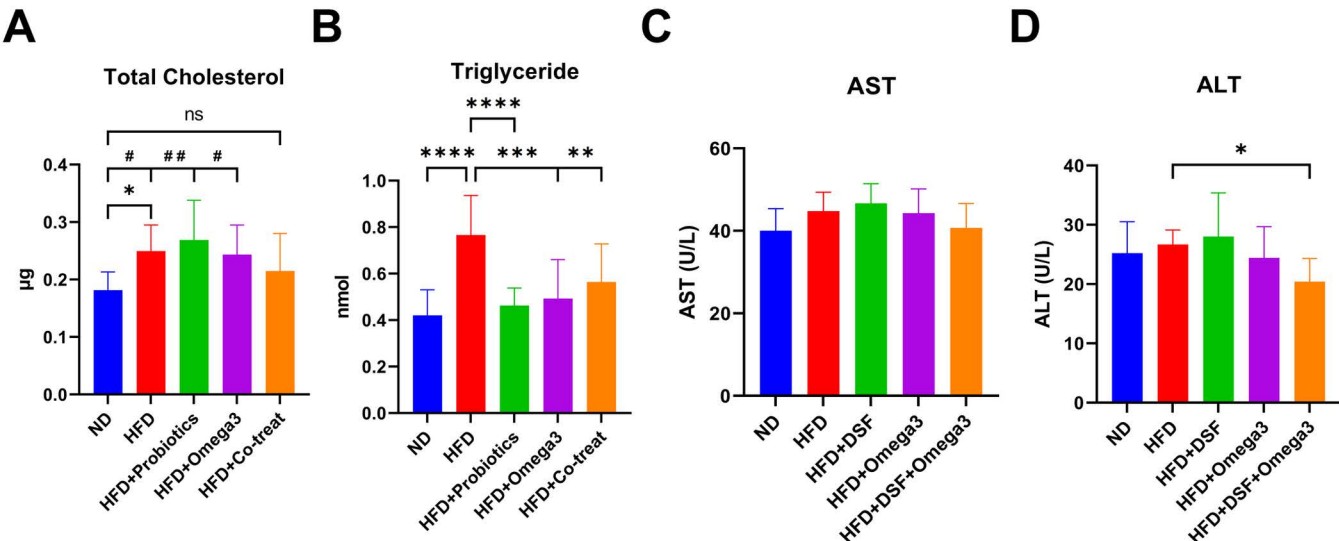

**Fig 2. Analysis of liver function bio-markers in mice co-administered with probiotics and omega-3.** (A-B) Total cholesterol and triglyceride levels in liver tissue, (C-D) Serum AST and ALT levels. Values represent the mean ± SD (n = 10 per group). Significance is indicated by *p < 0.05, **p < 0.01, ***p < 0.001, and ****p < 0.0001 as compared with the HFD group, and # p < 0.05, and ## p < 0.01 as compared with ND group.

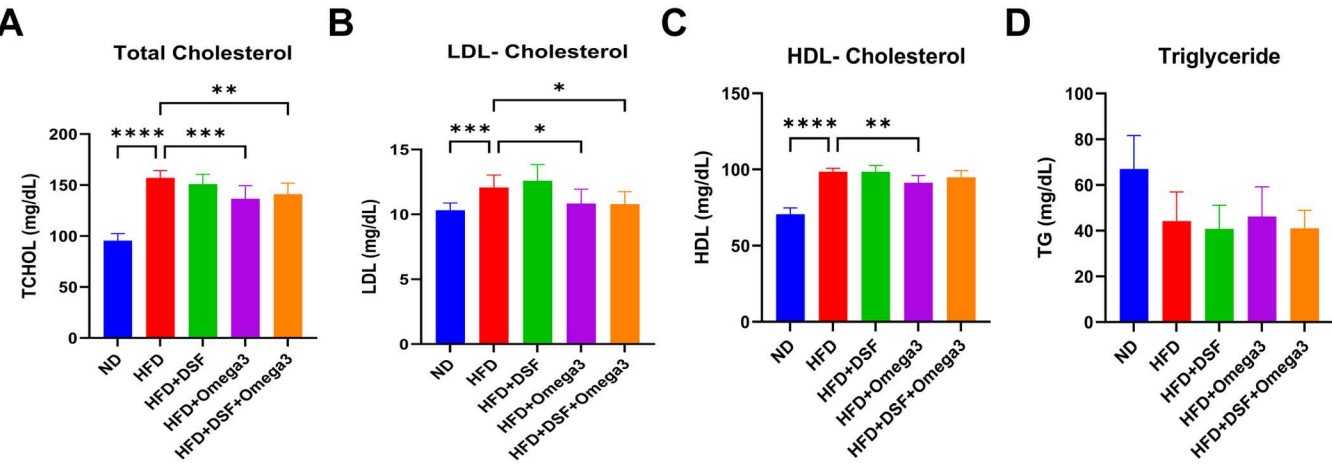

**Fig 3. Obesity-related bio-markers in plasma of mice co-administered with probiotics and omega-3.** Graphs showing results of quantitative analysis of (A) total cholesterol, (B-C) LDL and HDL, and (D) triglyceride levels. Values represent the mean ± SD (n = 10 per group). Significance is indicated by *p < 0.05, **p < 0.01, ***p < 0.001, and ****p < 0.0001 as compared with the HFD group.

a statistically significant difference was observed compared to the HFD group (Fig 3B). However, we found no significant differences between the HFD fed groups in plasma HDL and triglyceride levels (Fig 3C-D).

## Probiotics and omega-3 combined treatment inhibited thrombosis

To evaluate whether the administration of probiotics and omega-3 affects blood coagulation, we measured prothrombin time (PT) and activated partial thromboplastin time (aPTT), which are standard indicators of clotting function. Prolonged PT and aPTT values suggest delayed blood coagulation. The HFD + DSF + Omega-3 group showed longer clotting times

(PT: 7.3 ± 0.55 sec; aPTT: 26.7 ± 0.69 sec) compared to the HFD group (PT: 7.0 ± 0.29 sec; aPTT: 25.1 ± 1.80 sec), with the difference in aPTT reaching statistical significance. These results indicate that co-administration of DSF and omega-3 may delay coagulation, potentially contributing to improved blood flow and a reduced risk of thrombotic complications associated with obesity (Fig 4). The anti-coagulation index was analyzed using the t-test of the Graphpad Prism10 program.

### *Lactobacillus* abundance

Under HFD (high-fat diet) conditions, the relative abundance of *Lactobacillus* was markedly elevated in the combined treatment group (HFD + DSF + Omega3) compared to all other groups. As shown in Fig 5A, the HFD + DSF + Omega3 group exhibited a significantly higher *Lactobacillus* proportion than the HFD control group (p < 0.05). This combined supplementation group also surpassed the HFD + DSF and HFD + Omega3 groups in *Lactobacillus* levels. In contrast, no significant differences were observed between the single-supplementation groups (HFD + DSF or HFD + Omega3) and the HFD-only group, indicating that *Lactobacillus* abundance was only significantly increased with the combined probiotic and omega-3 treatment.

### Alpha diversity

Alpha diversity of the gut microbiota was significantly increased only in the combined supplementation group. Fig 5B showed that the HFD + DSF + Omega3 group had a higher alpha diversity (Shannon index) compared to the HFD group (p < 0.05). This increase in alpha diversity was unique to the combined treatment; neither DSF nor omega-3 alone produced a statistically significant change. There were no significant differences in alpha diversity between the HFD control, HFD + DSF, and HFD + Omega3 groups (p > 0.05), underscoring that the sole statistically significant gain in microbial diversity occurred in the presence of both probiotic and omega-3 together.

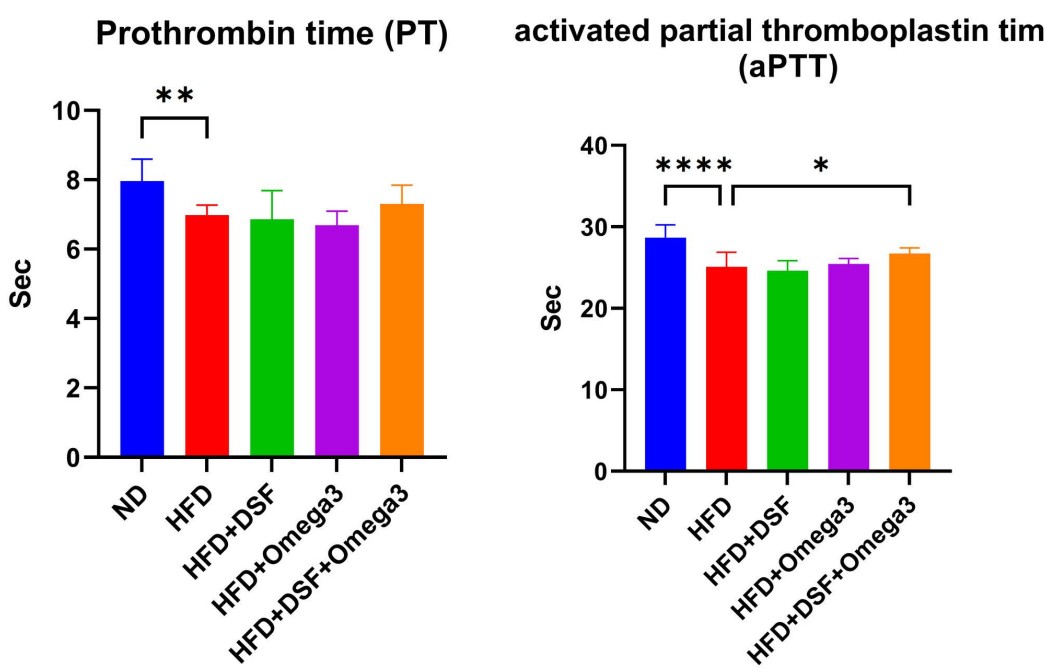

**Fig 4. Anticoagulation analysis in the plasma of probiotic-omega3 co-administered mice.** Graphs showing results of analysis of Assessment of prothrombin time (PT) and activated partial thromboplastin time (aPTT). Values represent the mean ± SD (n = 10 per group). Significance is indicated by *p < 0.05, **p < 0.01, ***p < 0.001, and ****p < 0.0001 as compared with the HFD group.

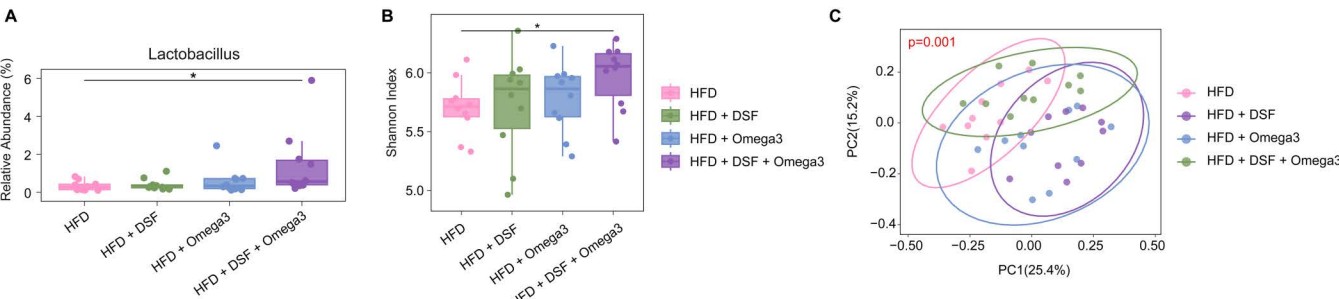

**Fig 5. Gut microbiota analysis in mice co-administered with probiotics and omega-3.** Graphs showing results of (A) *Lactobacillus* abundance, (B) alpha diversity (Shannon index), and (C) beta diversity analysis in cecal microbiota of mice administered high-fat diet (HFD), HFD+DSF, HFD+Omega3, or HFD+DSF+Omega3. *Lactobacillus* abundance and alpha diversity index were significantly increased in the HFD+DSF+Omega3 group compared to other groups. Beta diversity (PCoA plot) showed distinct microbial clustering between groups, with a significant difference in microbial community composition (p=0.001). Pairwise PERMANOVA results are described in the main text. Values represent the mean±SD (n=10 per group). Significance is indicated by *p<0.05, **p<0.01 compared with the HFD group.

## Beta diversity

Beta diversity was calculated using Bray-Curtis dissimilarity metric and visualized through principal coordinate analysis (PCoA). Beta diversity analysis revealed distinct clustering of microbial communities among the different treatment groups. Fig 5C presents a PCoA plot of beta diversity, where samples segregated by group. The HFD+DSF + Omega3 samples clustered separately from those of the HFD control and the single-supplement groups, reflecting a pronounced shift in community composition with combined treatment. Overall, the spatial separation of clusters for HFD, HFD+DSF, HFD + Omega3, and HFD+DSF + Omega3 was evident, and a permutational multivariate analysis of variance confirmed that the differences in microbial community structure among all groups were statistically significant (p=0.001). In addition, pairwise PERMANOVA revealed statistically significant differences between the HFD group and the other groups: HFD vs. HFP (p=0.013); HFD vs. HFO (p=0.002); HFD vs. HFC (p=0.001). These results further support the observation that co-administration of DSF and Omega-3 leads to the most distinct alteration in the gut microbiota composition.

## Discussion

Obesity is associated with various diseases, particularly affecting diseases such as CVD and metabolic dysfunction-associated steatotic liver disease (MASLD) due to the accumulation of cholesterol and triglycerides in the blood [25]. In addition, elevated levels of cholesterol and triglycerides in the blood can lead to the accumulation of lipids within liver cells, which contributes to liver cell injury. This injury results in the release of liver enzymes such as AST and ALT into the bloodstream, thereby increasing their serum levels and serving as biomarkers of liver damage [26–28]. Omega-3 is known as dietary supplement that helps inhibit the accumulation of triglycerides in the blood by inhibiting the synthesis of tri-glycerides in the liver [29,30]. They are also known to prevent atherosclerosis by inhibiting the generation of oxidized LDL, increasing HDL levels, and decreasing LDL levels in the blood [29,30]. Furthermore, lipids in the blood can cause cardio-vascular disease by promoting blood coagulation when the concentration is high [31–33]. In this study, high-fat diet obese male mice model showed a decrease in liver triglycerides and a decrease in serum cholesterol concentration by administering omega-3 and probiotics. In addition, the aPTT, a blood coagulation test, showed a combined effect in the group receiving combined administration of omega-3 and probiotics, indicating improved blood coagulation time. While omega-3 is known to reduce lipid levels such as cholesterol in the blood and liver tissue, the enhanced efficacy observed in the combination group suggests that co-administration with probiotics may have improved omega-3 bioavailability, leading to greater physiological benefits.

Obesity caused by a high-fat diet leads to fatty liver due to the accumulation of lipids such as cholesterol and tri-glycerides in the liver, and liver enzymes such as AST and ALT are released into the blood due to apoptosis of liver cells, so they are used as biomarkers of liver damage [34–36]. In this study, high-fat diet increased the accumulation of choles-terol and triglycerides in the liver tissue, while the individual administration of probiotics and omega-3 did not show any effect, while the combined administration of omega-3 and probiotics showed a combined effect by showing a total choles-terol content similar to that of the normal diet group. Although a statistically significant combined effect of the combined administration was not confirmed for hepatic triglyceride levels, treatment with probiotics, omega-3, and their combination each demonstrated a trend toward improvement compared to the high-fat diet group. Furthermore, it was found that the intake of probiotics through farnesoid X receptor (FXR), bile acid receptor, and KO mice induced metabolic dysfunction-associated steatohepatitis (MASH) through anti-inflammatory effects by removing liver lymphocyte infiltration and reducing liver fat content [37]. In this study, omega-3 intake significantly reduced ALT levels, and a greater reduction was observed when co-administered with probiotics, suggesting a potential combined effect. Although changes in AST levels were not statistically significant, a decreasing trend was noted in the combination treatment group. While these results do not pro-vide definitive evidence of improved AST levels, the observed trends suggest that combined administration of probiotics and omega-3 may contribute to alleviating liver injury, potentially through the anti-inflammatory properties of probiotics and a reduction in hepatic lipid accumulation.

The combined effect of co-administration of probiotics and omega-3 fatty acids in high-fat diet-induced obesity male mice has demonstrated promising outcomes in mitigating obesity-related metabolic dysfunctions. Specifically, the combi-nation treatment significantly reduced body weight gain, improved liver function by lowering total cholesterol and ALT lev-els, and positively influenced blood lipid profiles by reducing LDL levels. Additionally, the study revealed that the combined administration of omega-3 and probiotics significantly prolonged aPTT, indicating a delay in blood coagulation. This sug-gests a potential benefit in reducing the risk of thrombus formation, as obesity is often associated with a hypercoagulable state. Therefore, the observed prolongation of coagulation time may reflect an improvement in obesity-related cardiovas-cular complications by mitigating excessive coagulation tendencies. In addition to the metabolic improvements, our find-ings show that the omega-3 + DSF co-treatment positively modulated the gut microbiota. The gut microbiota has emerged as a key regulator of host metabolic health, influencing energy harvest, inflammation, and insulin sensitivity [38]. Mounting evidence indicates that an imbalanced microbiome (dysbiosis) is associated with metabolic disorders, whereas a diverse and balanced gut microbiota supports metabolic homeostasis [38]. In particular, individuals with obesity and related met-abolic abnormalities often exhibit reduced gut bacterial α-diversity (shannon index), and low microbial diversity has been linked to outcomes like increased adiposity and insulin resistance [39]. In our study, the combined omega-3 + DSF therapy led to a statistically significant increase in gut bacterial α-diversity, as well as a significant enrichment of the beneficial genus *Lactobacillus* in the fecal microbiota, compared to untreated high-fat diet controls. Beta diversity analysis further confirmed that the overall microbial community structure was markedly altered by the combined treatment, distinct from that of controls or single-treatment groups. Notably, omega-3 fatty acids may themselves exert prebiotic-like effects on the microbiome; omega-3 supplementation has been reported to promote the growth of beneficial gut bacteria and increase microbial diversity [40]. For example, omega-3 polyunsaturated fatty acids can enrich short-chain fatty acid-producing taxa and reduce potentially harmful microbes, thereby favorably shifting gut community composition [40]. The probiotic DSF likely contributed by directly introducing and supporting probiotic strains such as *Lactobacillus*. *Lactobacillus* species are well-known beneficial microbes that can improve gut health and have been associated with better metabolic outcomes [41]. Therefore, the observed rise in α-diversity alongside higher *Lactobacillus* abundance in the omega-3 + DSF group indicates a robust enhancement of gut ecosystem health. Although we did not directly examine functional metabolites in this study, the substantial microbiome changes suggest that combined omega-3 and probiotic therapy can beneficially reshape the gut environment. This gut microbiota modulation is a novel finding that adds an important dimension to the overall benefits of the combined treatment. These findings suggest that combined treatment with probiotics and omega-3

not only enhances the absorption of omega-3 but also exhibits a cooperative effect in improving metabolic health, particularly in obese males. This study was conducted exclusively in male mice, which limits the generalizability of the findings to both sexes. As metabolic responses can differ significantly between sexes due to hormonal and physiological factors, future research including both male and female animals is necessary to validate and extend these findings. Future studies could further investigate the mechanistic pathways involved and explore the long-term clinical applications of these findings, with careful consideration of sex differences in human populations.

## Supporting information

**S1 File. Raw data of experimental results.**
(XLSX)

## Acknowledgments

We appreciate the technical support of Minkyung Sung of Daegu-Gyeongbuk Medical Innovation Foundation (K-MEDI hub).

## Author contributions

**Conceptualization:** Tae-Jun Kwon.

**Data curation:** Kwang Hee Son, Tae-Jun Kwon.

**Formal analysis:** Jeongwook Lee, Hee-Young Yang.

**Funding acquisition:** Tae-Jun Kwon.

**Investigation:** Kwang Hee Son, Jeongwook Lee, Hee-Young Yang, Jong Seo Lee.

**Methodology:** Kwang Hee Son, Jeongwook Lee, Hee-Young Yang, Yerim Heo.

**Supervision:** Dong-Seok Lee, Kum-Joo Shin.

**Validation:** Jong Seo Lee.

**Writing – original draft:** Kum-Joo Shin, Tae-Jun Kwon.

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
