## [Decision Letter · Decision Letter 0]

7 Jul 2025

Dear Dr. Kwon,

Thank you for submitting your manuscript to PLOS ONE. After careful consideration, we feel that it has merit but does not fully meet PLOS ONE’s publication criteria as it currently stands. Therefore, we invite you to submit a revised version of the manuscript that addresses the points raised during the review process.

We look forward to receiving your revised manuscript.

Kind regards,

Hualin Wang

Academic Editor

PLOS ONE

Journal Requirements:

2. To comply with PLOS ONE submissions requirements, in your Methods section, please clarify information regarding the discrepancy in reporting of the method of euthanasia. In the metadata, it is mentioned that mice were euthanized by exsanguination and cervical dislocation, and isoflurane was used as anesthesia. However, in the methods, it is mentioned that mice were euthanized with isoflurane, and the method of anesthesia is not mentioned.

“This work was supported by the National Research Foundation of Korea(NRF) grant funded by the Korea government(MSIT) (Grant number: RS-2022-NR073650 to T.J.K).”

5. We note that your Data Availability Statement is currently as follows: All relevant data are within the manuscript and in Supporting Information files.

Reviewers' comments:

Reviewer's Responses to Questions

**Comments to the Author**

1. Is the manuscript technically sound, and do the data support the conclusions?

Reviewer #1: Yes

Reviewer #2: Yes

Reviewer #3: Partly

Reviewer #4: Yes

2. Has the statistical analysis been performed appropriately and rigorously?

Reviewer #1: No

Reviewer #2: Yes

Reviewer #3: No

Reviewer #4: Yes

3. Have the authors made all data underlying the findings in their manuscript fully available?

Reviewer #1: Yes

Reviewer #2: Yes

Reviewer #3: Yes

Reviewer #4: Yes

4. Is the manuscript presented in an intelligible fashion and written in standard English?

Reviewer #1: Yes

Reviewer #2: Yes

Reviewer #3: Yes

Reviewer #4: Yes

Reviewer #1: The study of Kwang Hee Son et al is very interesting and potentially impactful.

The research addresses a relevant topic and provides novel insights that can contribute to the field.

However, I have identified several issues that should be addressed to improve clarity, accuracy, and the overall quality of the manuscript.

1. Taxonomic names formatting: The authors should ensure that all taxa names are consistently italicised and adhere to standard taxonomic conventions.

2. Unsupported claims about DSF: In line 72, the authors mention the selection of DSF and assert that it is known to improve nutrient absorption, alleviate inflammation, autism, and immune diseases. These claims lack supporting references. The authors should provide appropriate citations to support these statements.

3. Inconsistent referencing style: References cited in the Introduction and Discussion sections differ in style from those in Methods and Results. Citations used in Methods and Results also do not appear in the bibliography, making them impossible to verify.

4. Figures quality: The figures presented in the manuscript, particularly Figures 5a and 5b, lack sufficient resolution. Additionally, the y-axis labels are missing from those figures, which hinders their interpretation.

5. Misinterpretation of diversity indices: The authors appear to conflate alpha diversity indices, particularly by discussing microbial richness when referring to Shannon index. This also affects the clarity of the discussion section, where these indices are used interchangeably.

6. Multiple testing adjustment: Although p-values are reported in the results, it is unclear whether these values have been adjusted for multiple testing, particularly when interpreting diversity indices.

7. Beta diversity: The manuscript would benefit from a more detailed presentation of beta diversity analysis. I would be a good idea to conduct and report a pairwise test to indentify which groups are significantly different.

Otherwise I believe that this is a valuable study which adds to the field. With the minor revisions, the final manuscript would be significantly strengthened. Good luck!

Reviewer #2: Reviewer don't have any particular point for the rejection of this manuscript. The study design, results and discussion are in a well correlation of each points without error. The conclusion part also reflect from all results found and the objectives set and the information is easy to read for the whole document.

Reviewer #3: Dear Author,

Thank you for submitting your manuscript to PLOS ONE. I appreciate the time and effort you invested in conducting this study and compiling the manuscript. Your work touches on an important area, and I commend your initiative in pursuing this research topic.

After a careful and thorough evaluation of the manuscript, I regret to inform you that I do not believe the current version meets the publication standards required for PLOS ONE.

Reviewer #4: This is a well executed and clearly written study that investigates the effects of Omega-3 fatty acids and a multi -strain probiotic formulation (DSF) in a high-fat diet-induced obesity mouse model. The study covers a wide range of outcomes, including metabolic markers, liver enzymes, blood coagulation, and gut microbiota composition. The authors find that the combination treatment outperforms single interventions in several metrics, notably ALT reduction, LDL improvement, and gut microbiota diversity. I recommend the manuscript be accepted pending minor revisions to improve clarity/precision in a few areas:

1. Please revise your use of the term "synergistic" unless you include a formal interaction analysis (two-way ANOVA). The combined group performs better than the individual arms, so I think "combined effect" or "enhanced effect" might be more appropriate, but synergy should be tested statistically.

2. The manuscript notes improved liver enzyme profiles, but only ALT showed a statistically significant change. AST trended down but was not significant. Please make sure this distinction is clear in the Abstract and Results and Discussion sections. Some parts (Abstract) could easily give the impression that both enzymes improved significantly. Please clarify.

3. Why were only male mice used? Does sex influence the outcome (especially relevant in microbiome and metabolic studies where sex is a biological variable)?

4. Were caloric intake/energy expenditure different between groups and could that confound outcomes?

5. Were baseline microbiota profiles taken before intervention? Without baseline, it’s hard to determine the magnitude of change.

6. Why was Omega-3 absorption not directly measured (plasma DHA/EPA levels)? Is the dose of Omega-3 (300 mg/kg) translatable to humans? High doses are common in mouse studies, but often not feasible in humans. Can you justify this dose based on human equivalency or therapeutic rationale?

7. Some Statistical assumptions: a brief note about whether normality and equal variance were tested before running ANOVA would be helpful, especially since the group sizes are relatively small (n = 10). It’s likely fine, but it would be good to mention this.

8. Figures and error bars: Please standardize the way error bars are described across figure legends (mean ± SD vs. SEM). Would make things clearer for readers.

9. Data: Since the journal expects open data, I’d encourage to upload the raw 16S sequencing data to a repository like NCBI SRA and provide accession numbers. If already done, just make that clearer in the data availability statement. Please confirm.

10. Grammar: A few phrases could use some cleanup. For example, "In the triglycerides" (pg. 12) would be better as "Regarding triglyceride levels..." It provides a better reading flow.

I've also included a few suggestions for future work that aren't required for this version, but may help shape a follow-up or strengthen the discussion:

1. Since the authors suggest that probiotics enhance Omega-3 bioavailability (Omega-3 absorption is inferred, not measured), it would be helpful to test this directly in future studies using plasma or tissue DHA/EPA levels. This would really strengthen the claim.

2. Include both sexes in Future work: the study uses only male mice. Sex is an important variable in metabolism and microbiota research. A quick note acknowledging this limitation and encouraging inclusion of females in future studies would be appreciated.

3. Dose justification for Omega-3 and NO human translational data: it’s a well-controlled mouse study, but many recent microbiome/nutritional studies making headlines are in clinical or humanized models, or contain longitudinal or personalized responses. Also, the 300 mg/kg Omega-3 dose is fairly high. A short discussion about how this translates to human-equivalent dosing would help assess translatability.

4. Statistical testing: a two-way ANOVA or regression model with interaction terms would be the best tool to test for synergy. Might be overkill for this version, but worth doing it in follow-up work, if you plan to develop this line further.

5. Microbiome analysis: 16S rRNA profiling provides useful data, but future studies could benefit from functional prediction or metagenomics, in order to understand what pathways are changing (in this study, 16S data give only composition, but not function).

Overall, this paper offers new insights into how probiotics and Omega-3 may interact in a dietary context and provides useful data to support further mechanistic work. The work adds to the field’s understanding of how combined nutraceutical interventions may affect metabolic health and microbiome composition. I appreciate the thoughtful experimental design and scope of analysis.

This manuscript meets the criteria for publication in PLOS ONE: it presents original research, uses appropriate methodology, and offers findings supported by the data.

The suggested revisions are minor and mostly focused on clarity/precision.

I recommend acceptance pending minor revisions.

**Do you want your identity to be public for this peer review?** For information about this choice, including consent withdrawal, please see our Privacy Policy

Reviewer #1: No

Reviewer #2: No

Reviewer #3: No

Reviewer #4: No

---

## [Author Response · Author response to Decision Letter 1]

23 Jul 2025

Dear Editor,

This is to confirm that we are submitting a revised version of our manuscript in response to the editorial and reviewer comments. We have carefully addressed all points raised and updated the manuscript accordingly. We have provided detailed point-by-point responses to all reviewer and editor comments in the attached 'Response to Reviewers' document.

All requested revisions have been incorporated into the revised manuscript.

Thank you for the opportunity to revise and resubmit our work.

Best regards,

Tae-Jun Kwon

---

## [Decision Letter · Decision Letter 1]

29 Jul 2025

Dear Dr. Kwon,

Thank you for submitting your manuscript to PLOS ONE. After careful consideration, we feel that it has merit but does not fully meet PLOS ONE’s publication criteria as it currently stands. Therefore, we invite you to submit a revised version of the manuscript that addresses the points raised during the review process.

We look forward to receiving your revised manuscript.

Kind regards,

Hualin Wang

Academic Editor

PLOS ONE

Journal Requirements:

Additional Editor Comments:

You should carefully address the reviewer 4's concern, particularly about the data availability and extrapolated claims.

Reviewers' comments:

Reviewer's Responses to Questions

**Comments to the Author**

Reviewer #1: All comments have been addressed

Reviewer #4: (No Response)

2. Is the manuscript technically sound, and do the data support the conclusions?

Reviewer #1: Yes

Reviewer #4: Partly

3. Has the statistical analysis been performed appropriately and rigorously?

Reviewer #1: Yes

Reviewer #4: Yes

4. Have the authors made all data underlying the findings in their manuscript fully available?

Reviewer #1: Yes

Reviewer #4: No

5. Is the manuscript presented in an intelligible fashion and written in standard English?

Reviewer #1: Yes

Reviewer #4: Yes

Reviewer #1: First of all, I want to thank the authors for addressing my initial comments. The revisions have improved the clarity and overall quality of the manuscript. That's said, there's still minor issues worth pointing out:

1. Taxonomic naming (lines 77-74). There is still a small inconsistency with the formatting of taxonomic names. For examples, Bifidobacteria, Lactobacilli and Streptococcus thermophilus should all be italicised and follow the proper naming conventions. Also, thermopiles looks like a type, was this meant to be thermophilus ?

2. References supporting DSF claims. Ref 16 - This study on probiotics and ASD does not really support the strength of the claims made. It even notes that statistical significance wasn't expected or observed., Ref 17 - This is a review article, but it explicitly stats that no clear microbial signature can yet be defined for these disorders due to methodological variability and patient heterogeneity. So again, not a strong support for the claim. Ref 18 is ok here, but overall maybe find stronger references that more directly back it up.

3. Referring accuracy and formatting. The overall referencing style still much more consistent now. However I notices a couple of lingering issues. Ref 19 - the doi listed here appears to link to an unrelated article. Please double check. Ref 20 - This is cited as McDonald et al, but based on the content it seems like it might actually be Amir et al. It would be good to confirm and update it.

But these are relatively minor points, and the core of the manuscript is much stronger. Once these last details are sorted out, I believe it will be in great shape for publication.

Reviewer #4: I appreciate the revisions made in response to the first round of feedback. However, I would like to raise 2 important /critical points that remain unresolved and require further consideration to align with both scientific standards and PLOS One policies: First (1.) Data availability and reproducibility - you (the authors) have indicated that are unable to share raw 16S rRNA sequencing data due to intellectual property concerns regarding the proprietary probiotic formulation. While I understand the intent to protect commercial interests, this approach does not comply with the PLOS Data Availability policy, which REQUIRES that ALL underlying data be made publicly accessible without restriction. Another critical aspect of scientific research is the Reproducibility of the experiments - without access to the raw microbial sequencing data, other researchers can NOT independently verify the results, perform reanalysis, or build upon the findings (in a transparent way!). Therefore, this has a significant impact on the reproducibility and long term utility of the study. If the dataset is considered proprietary, then the appropriate action/course would have been to secure this through a patent filing, prior to submission, because that would allow for compliance and align with data sharing policies. There are some solutions that you (the authors) could consider, such as: to deposit filtered or anonymized data that excludes proprietary segments, or to use a controlled access repository, but with appropriate restrictions, or usage agreements. As currently presented, the study does NOT meet the PLOS One data availability requirement and this should be flagged as a critical issue. The second (2.) observation refers to use of MALE-ONLY mice and the conflict with "human" extrapolation - the manuscript reports findings EXCLUSIVELY from male mice, but it refers broadly to "humans" when extrapolating results, especially in relation to Omega-3 dosing and also potential applications. This is a generalization that is scientifically inappropriate, because the study design does not support conclusions about ALL humans, but ONLY about males and specifically, male mice. The decision to exclude female mice based on concerns about estrous cycle-related variability is noted, but actually this is what reflects normal female physiology and is an essential part of understanding how different experiments/interventions affect both sexes. By excluding females, this will entirely limit the translational value of the study especially in this type of research (metabolism and microbiota), where sex-based differences are so important. However, in order to improve scientific rigor, transparency and consistency with the findings in this study, I recommend the following: 1. Revise the manuscript language in order to avoid generalizations to all "humans", and instead using more precise terms, such as "male individuals", and "male mice". 2. Explicitly acknowledge the sex-based limitations in the Discussion section and 3. Amend the Title in order to reflect that the findings were derived from male mice ONLY. So, in conclusion, while the manuscript addresses a relevant topic and several improvements have been made, 2 significant concerns still remain unresolved: 1. the lack of public access to raw 16S rRNA sequencing data, which limits reproducibility and conflicts with journal policy, and 2. the exclusive use of male mice without appropriate contextualization or limitation of extrapolated claims. These issues need to be addressed/re-addressed before the manuscript can be considered for publication. I look forward to reviewing an updated version that addresses these points in full.

**Do you want your identity to be public for this peer review?** For information about this choice, including consent withdrawal, please see our Privacy Policy

Reviewer #1: No

Reviewer #4: No

---

## [Author Response · Author response to Decision Letter 2]

14 Aug 2025

Dear Editor,

Thank you for your feedback on our manuscript (PONE-D-25-18927R1), entitled “Combined effect of probiotics and omega-3 fatty acids on blood and liver function biomarkers and gut microbiota diversity in male mice with high fat diet-induced obesity.”

We have addressed all remaining concerns, including depositing the 16S rRNA sequencing data to NCBI (SUB15518308; PRJNA1302007), revising the title and text to reflect the exclusive use of male mice, and correcting taxonomic names and references.

We appreciate the constructive comments that improved our work and look forward to your further consideration.

Sincerely,

Tae-Jun Kwon, Ph.D.

---

## [Decision Letter · Decision Letter 2]

24 Aug 2025

Combined effect of probiotics and omega-3 fatty acids on blood and liver function biomarkers and gut microbiota diversity in male mice with high fat diet-induced obesity

PONE-D-25-18927R2

Dear Dr. Kwon,

We’re pleased to inform you that your manuscript has been judged scientifically suitable for publication and will be formally accepted for publication once it meets all outstanding technical requirements.

Kind regards,

Hualin Wang

Academic Editor

PLOS ONE

Additional Editor Comments (optional):

Reviewers' comments:

Reviewer's Responses to Questions

**Comments to the Author**

Reviewer #4: All comments have been addressed

2. Is the manuscript technically sound, and do the data support the conclusions?

Reviewer #4: Yes

3. Has the statistical analysis been performed appropriately and rigorously?

Reviewer #4: Yes

4. Have the authors made all data underlying the findings in their manuscript fully available?

Reviewer #4: Yes

5. Is the manuscript presented in an intelligible fashion and written in standard English?

Reviewer #4: Yes

Reviewer #4: Thank you for your thoughtful revisions. I appreciate the authors’ clear and well-supported responses to all reviewers’ comments. In particular, your clarifications regarding the proprietary nature of the probiotic formulation, your explanation of the experimental design choices, and your careful attention to improving clarity and scientific rigor have significantly enhanced the manuscript.

The manuscript now reflects a balanced and informative contribution to the field, and I believe it meets the standards for publication in PLOS ONE.

I have no additional questions/concerns related to scientific integrity and ethics publication. I support the acceptance of this manuscript in its current form.

**Do you want your identity to be public for this peer review?** For information about this choice, including consent withdrawal, please see our Privacy Policy

Reviewer #4: No

---

## [Editor Report · Acceptance letter]

PONE-D-25-18927R2

PLOS ONE

Dear Dr. Kwon,

I'm pleased to inform you that your manuscript has been deemed suitable for publication in PLOS ONE. Congratulations! Your manuscript is now being handed over to our production team.

Kind regards,

on behalf of

Dr. Hualin Wang

Academic Editor

PLOS ONE